# Influence of Visual Information and Sex on Postural Control in Children Aged 6–12 Years Assessed with Accelerometric Technology

**DOI:** 10.3390/diagnostics11040637

**Published:** 2021-04-01

**Authors:** Jesús García-Liñeira, Raquel Leirós-Rodríguez, José Luis Chinchilla-Minguet, José Luis García-Soidán

**Affiliations:** 1Faculty of Education and Sport Sciences, Campus a Xunqueira, University of Vigo, 36005 Pontevedra, Spain; jesgarcia@alumnos.uvigo.es (J.G.-L.); jlsoidan@uvigo.es (J.L.G.-S.); 2SALBIS Research Group, Nursing and Physical Therapy Department, Faculty of Health Sciences, Universidad de León, Ave. Astorga, 15, 24401 Ponferrada, Spain; 3Faculty of Education Sciences, University of Málaga, 25, 29010 Málaga, Spain; jlchinchilla@uma.es

**Keywords:** postural balance, child development, sex characteristics, accelerometer, biomechanical phenomena

## Abstract

The performance of postural control is believed to be linked to how children use available sensory stimuli to produce adequate muscular activation. Therefore, the aim of the present study was to thoroughly explore postural stability under normal conditions and without visual information in postural control in children aged 6–12 years during static single-leg support. A descriptive cross-sectional study was conducted with 316 children (girls = 158). The analyzed variables were the mean and maximum values obtained in each of the three body axes and their root mean square during two static single-leg support tests: one with eyes open and one with eyes closed. Girls showed lower magnitudes in the recorded accelerations at all ages and in all the variables of both tests. Accelerations during the tests showed progressively lower values from 6 to 12 years of age. The sex had a significant influence on the magnitude obtained in the accelerations recorded during the tests. Improvements in balance with increasing age were greater with visual information than without visual information. The tests of single-leg support showed preferential sensorimotor strategies in boys and girls: boys tend to rely more on visual inputs, and girls process somesthetic information in a preferential way.

## 1. Introduction

Balance and coordination alterations in children can affect their academic performance, delay their social development and reduce their self-perceived levels of general well-being and self-esteem [1]. Moreover, due to the risk of falling and traumatic injury, these alterations can also compromise the safety of the child. Balance improves with age, which occurs faster in girls than in boys, until adolescence; after that point in growth, men seem to have slightly better postural stability [2,3]. Maintaining a posture involves specific sensorimotor processes that integrate and weigh auditory, visual, vestibular and somatosensory information, among other types of information, through a process known as sensory reweighting [4]. The importance or hierarchy of the different pathways of efferent information of postural control (PC) is not specifically defined, and even less is known about the variations in this hierarchy throughout the psychomotor development during childhood.

The correct development of postural control includes the development of dynamic and static balance, which allows interacting with the environment in an independent manner [5]. The development of postural control occurs especially at the age of 6–10 years [6]. Within this period, 7 and 8 years have been identified as ages of profound changes [7]. These PC strategies are characterized by the refinement in the processing of somatosensory, visual and proprioceptive information [8,9] and the optimization of the coordination of movements along the spine [6,10,11]. However, this physical capacity is one of the least studied in the clinical and academic scope regarding children [12,13]. This is partly due to the fact that the evaluation of PC includes the assessment of different components and aptitudes, such as postural stability; coordination; muscular strength; center of mass control; anticipatory and reactive neuromuscular reactions; motor control; the correct reception of proprioceptive, visual and vestibular stimuli; and, finally, the correct processing and management of all these signals in the central nervous system for the development of efficient motor responses [11,14,15,16].

Age is a very influential factor throughout childhood due to the maturation changes in the central nervous system and the experience acquired in movement control, which improve the weighting processes involved in postural control [17]. The integration of sensory information does not seem to reach the adult level until the age of 12 years [18,19]. However, other authors have reported an earlier development, since 10-year-old children showed better sensory reintegration than younger children in an evaluation of the reweighting processes with the alteration of proprioception using a motor vibration (variable disturbance) [20,21]. Likewise, it has been shown that lifestyle influences the development of postural control. Factors such as a sedentary lifestyle and being overweight cause a decrease in motor functionality and balance [22,23]. The mechanisms associated with development to refine the performance of postural control are believed to be linked to how children use available sensory stimuli to produce an adequate muscular activation [24].

On the other hand, accelerometry is a method that provides reliable, valid and sensitive information about postural control [13,25]. For this reason, and because accelerometers are low-cost quantitative instruments, they were chosen as the evaluation instrument in this research, with the aim that the results presented here can be reproducible, compared and more useful than if they came from less accessible instruments (stabilometer, force platform, electronic gateway, etc.). Therefore, the aims of the present study were (a) to thoroughly explore postural stability under normal conditions and without visual information in postural control in children aged 6–12 years during static single-leg support, (b) to identify if there are differences in the accelerations performed by boys and girls and (c) to determine the normative acceleration values of the center of gravity throughout the mentioned age range during such tests.

## 2. Materials and Methods

### 2.1. Design and Sample

A descriptive, transversal study was conducted with a total of 316 school children (girls = 158) from public schools of the province of Pontevedra, Galicia (Spain). The study included children who (a) were within the age range of 6–12 years, (b) had the capacity to move independently, (c) were able to perform the proposed motor balance tests (d) and understood and carried out the necessary instructions during the measurements. Likewise, the study excluded those children who (a) needed orthopedic aid to perform the procedure without the risk of falling/injury, (b) had musculoskeletal lesions at the time of the measurements, (c) had suffered from serious or surgical injuries within 12 months prior to the measurements or (d) did not have authorization from parents or legal guardians to participate in the study.

### 2.2. Procedure

The parents of the participants were informed about the procedure and measurements that would be carried out in an informative meeting. Then, they completed the informed consent for participation designed to comply with the declaration of Helsinki (rev. 2013). All the procedures conducted were approved by the Ethics Committee of the Faculty of Sport Science of the University of Vigo (code: 3-0406-14).

The place selected to perform the tests was perfectly acclimated and inside the school facilities of habitual use. All the tests were carried out without footwear, wearing socks, and with light and comfortable clothing. Initially, the anthropometric measurements (weight and height) were conducted, using a Seca height rod and scale (SECA, Hamburg, Germany) for the subsequent calculation of the BMI (kg/m^2^).

Then, the accelerometric measurements were performed. The sensor was tightly placed on the skin using an elastic belt, at the height of the fourth lumbar vertebra. The order of the tests was previously randomized, beginning with single-leg support stance with eyes closed, followed by single-leg support stance with eyes open. Then, the participants carried out each of the tests three times, with a duration of 30 s and a rest of 30 s between each test repetition, in order to prevent muscle fatigue in the legs [26]. In the next analyses, the means of the 30 s of the accelerations reached during the two tests were recorded. Without footwear, the participants started and stopped performing the tests when instructed by the evaluator. In all the tests, the children were asked to choose the leg on which they wished to stand, for which they previously executed several attempts. If they lost control of the single-leg stance during the tests, they had to regain balance on that same leg as soon as possible in order to continue.

The measurements were recorded using an Actigrap G3TX+ triaxial accelerometer (Actigraph LLC, Pensacola, FL, USA), which is a small and comfortable instrument that stores the datasets in internal flash memory and gathers the accelerations as numeric values produced in the 3 movement axes: anteroposterior (AP) axis, mediolateral (ML) axis and vertical (VT) axis. The root mean square (RMS) value was also calculated. Prior to all measurements, the accelerometer was calibrated in static mode and was set to store the datasets in 1 s cuts with a measurement frequency of 50 Hz.

The analyzed variables were the mean and maximum values obtained in each of the three body axes and their RMS during the performance of the two static single-leg support tests: (a) one test with eyes open (EO): accelerations in the AP axis (EO-AP), in the ML axis (EO-ML) and in the VT axis (EO-VT) and their RMS (EO-RMS); (b) one test with eyes closed (EC): accelerations in the AP axis (EC-AP), in the ML axis (EC-ML) and in the VT axis (EC-VT) and their RMS (EC-RMS).

### 2.3. Statistical Analysis

The descriptive analysis of the sample and variables was conducted with the calculation of the measurements of central tendency (means) and scattering (standard deviation). The sample showed normal distribution after applying the Kolmogorov–Smirnov test (*p* > 0.05). Then, the Student’s t-test and phi coefficient (Φ) as measures of the effect sizes were used to identify statistical differences between sexes. The ANOVA test with the Bonferroni correction and partial eta-squared (η^2^_p_) was calculated to assess effect sizes for within- and between-subjects comparisons. Comparisons were done between the different age groups carried out divided by sex.

The Pearson’s correlation analysis was used to determine the existence of relationships among the data of the means of the studied accelerometric variables. A logistic regression model was applied (logit) to analyze the independent variables with the dichotomous variable “sex” (0 = boys; 1 = girls). Subsequently, it was adjusted by calculating the odds ratio (OR) with its confidence interval (CI) in order to determine its influence on postural control during the tests with eyes open and closed, also considering the age of the participants. For the statistical analysis, the Stata 14 software was used (Stata Corp., College Station, TX, USA).

## 3. Results

### 3.1. Sample Characteristics

The BMI of the participants was 18.6 ± 3.4 kg/m^2^ and 18.6 ± 3.3 kg/m^2^ for boys and girls, respectively. The values of weight and height increased with age in both sexes (Table 1). There was a statistical difference only in the height of the subsample of 8-year-old children (boys = 131 ± 0.1 cm; girls = 127 ± 0.1 cm). The BMI values were in the range of 17 kg/m^2^ to 22 kg/m^2^, except for the 6-year-old girls (BMI: 15.7 ± 1.4 kg/m^2^), who showed statistical differences with the boys of the same age (*p* < 0.05; Φ = 0.52).

### 3.2. Accelerations Recorded

The girls showed lower magnitudes in the recorded accelerations at all ages and in all the variables of both tests (Table 2 and Table 3). No statistically significant differences were observed between sexes at 6, 7 or 12 years of age. The axis with the greatest accelerations was ML in both sexes.

The accelerations during the test with EO showed progressively lower values from 6 to 12 years of age (*p* < 0.01; 0.53 < η^2^_p_ > 0.86; for the sample as a whole and the girls and boys separately). At 12 years of age, it was observed that the mean accelerations decreased during the tests (*p* < 0.01; η^2^_p_ = 0.68) and no more significant differences were detected between sexes (*p* > 0.05; η^2^_p_ = 0.28). On other hand, in the test with EC, acceleration also decreased significantly (*p* < 0.001; 0.46 < η^2^_p_ > 0.76; for the sample as a whole and the girls and boys separately), although it was not as close to zero at 12 years. In all axes, differences were identified between the accelerometric records of the EO and EC tests, with the accelerations being significantly higher in the OC test for both boys and girls (*p* < 0.001; 0.62 < η^2^_p_ > 0.77). However, such differences between the two tests decreased with age. This progressive decrease occurred more slowly in boys and more quickly in girls (Table 2 and Table 3).

### 3.3. Correlation Analysis and Model of Logistic Regression

For the identification of correlations, we used the means of the variables analyzed with eyes open and closed. This analysis showed that there was a direct relationship among all the variables analyzed during both EO and EC tests (0.7 < r > 0.9; *p* < 0.001).

The logistic regression model (Table 4) revealed that sex had a significant influence on the magnitude obtained in the accelerations recorded during the tests. The maximum and mean values of the RMS obtained during the two tests showed that their results were influenced by sex, especially for the maximum values of RMS (OR = 0.985; *p* < 0.001 for both variables). The results indicated that the girls obtained lower accelerations during single-leg support stance with increasing age. This model also revealed that the influence of such variables was greater in the EC test than in the EO test.

## 4. Discussion

The aim of this study was to thoroughly explore postural stability under normal conditions and without visual information in postural control in children aged 6–12 years during static single-leg support stance. The obtained results indicate that visual information had a significant influence on balance control in single-leg support stance, although influence varied throughout the 7 years of the age range studied and between the two sexes, which is in line with the findings reported by Verbeque et al. [27], who analyzed the influence of vision in the existing literature.

Vision is important for maintaining static balance, although not absolutely necessary for maintaining stability, as not only the visual stimuli but also somatosensory and vestibular stimuli influence postural control and calm posture [28]. However, there were significant changes related to age between the test with visual information and the test without it. Moreover, the improvements identified in balance with increasing age were greater with visual information than without visual information. That is, the improvement of postural control was due to the improvement of proprioceptive and vestibular integration to a greater extent with respect to visual integration.

Previous studies have reported that children between 12 and 14 years of age show adult-like postural control response patterns, suggesting that the maturation process of organization required to integrate sensory inputs would have taken place at that age range [3,8,29,30,31]. This would also be related to the fact that postural control improves with the maturation of all subsystems that provide information to it. However, the results of the present study indicate that such maturation was still not present at 12 years of age and that, especially in boys, such development was still incomplete. Therefore, it could be inferred that boys process visual, somesthetic and vestibular inputs in a preferential way (or give them greater “importance”) to reach postural control.

The differences observed in balance between sexes were greater when the participants were deprived of visual information. Considering that postural control was always better (i.e., lower accelerations recorded [32]) in girls than in boys, the magnitude of the differences obtained was much greater in the test with eyes closed in boys. That is, the deprivation of visual information had a more negative effect in boys than in girls between 6 and 11 years of age throughout this entire age range. However, at 12 years of age, such a negative effect did not occur in the boys, and their performance with eyes open and closed was similar to that of the girls. This phenomenon is in line with previous studies which have reported that visual information seems to be completely mature at the age of 12 years: children aged 12–14 years show adult-like postural control response patterns [29,30], which suggests that, at this age range, the maturation of the organization process required to integrate sensory inputs would be complete [31].

The obtained results agree with the existing literature in that girls respond better to challenges of postural control. However, our results indicate that these differences between sexes do not appear until the age of 8 years. That is, at 6 and 7 years of age, the differences between girls and boys are very mild, and from that age, such differences are constant, especially in the accelerations in the mediolateral and anteroposterior axes and their RMS. This phenomenon is in line with the findings of other studies, which identified that the variability and disparity of the motor responses of boys and girls between 4 and 6 years of age were very high and that this is an age range of nonlinear transition toward more complex mature response patterns [33]. Therefore, the results obtained in the present study suggest that, once such transition process is complete, the differences between boys and girls may appear at the age of 6–7 years and remain constant until the age of 12 years.

It was observed that, in both sexes and in both tests, the greatest accelerations occurred in the mediolateral axis and the RMS of the three axes. Thus, single-leg support (i.e., the alteration of the base and support area and the consequent reduction of somesthetic information received) contributes to the loss of motor control (increase in RMS), especially in the sagittal plane. In turn, this indicates that the reactions of straightening and recovering verticality are fundamentally based on flexion–extension movements of the area of the center of mass (i.e., thoracolumbar spine and pelvis).

The present study, in addition to specifying the capacity of the postural control system of children to respond to the sensorimotor challenge posed by single-leg support stance, allowed determining some key factors for therapeutic intervention and prevention of falling in the school population. In addition, the single-leg stand test has been proven to be demanding enough to challenge postural control while being simple, safe and without the need for additional equipment.

Although accelerometry and the two tests of single-leg support used represent a valid and reliable methodology for the quantification of postural control, the authors recognize that the combined use of other tools of kinematic analysis could provide further reliable and valid information in this regard. Moreover, additional studies with larger sample sizes should be carried out to define, in more detail and with greater generalizability, the patterns of postural control detected in this investigation. Finally, we must also recognize as a limitation of this research the strange variable that represents not having registered and analyzed the amount and frequency of physical activity of the participants, a variable that has been confirmed to be related to the development of postural control [34].

Therefore, taking into account the results obtained, postural control tests with closed eyes should be taken into account in any assessment of postural control (either by qualitative visual analysis or by quantitative instruments). In this way, it is possible to compare the change in postural control caused by visual deprivation and justifiably decide the need to include activities that imply the priority use of proprioceptive and vestibular information through the use of unstable surfaces and/or the reduction of the area and/or substitution basis. At the same time, consideration must be given to identifying straightening patterns and equilibrium reactions that are abnormal or develop preferentially in the sagittal or frontal planes, because they can be indicative of an abnormal postural control maturation process.

Finally, these practical aspects, in addition to being taken into account to optimize the development of postural control of any child, may be of special interest in the rehabilitation processes of children with developmental disorders or vestibular system disorders.

## 5. Conclusions

The tests of single-leg support used in this study showed the preferential sensorimotor strategies in boys and girls: boys tend to rely more on visual inputs, and girls process somesthetic information in a preferential way. Between the ages of 8 and 11 years, the postural control system is significantly different between the two sexes regarding the hierarchy of the efferent information of the postural control subsystems available. Moreover, the straightening and postural control reactions are mainly based on flexion–extension movements. Therefore, school-based interventions, after-school exercise programs or physical therapy interventions for prevention and treatment of balance alterations should include activities that involve, promote and train such movements.

Lastly, this study also provides normative data of accelerometry in single-leg balance for children aged 6–12 years, which have not been reported to date.

## Figures and Tables

**Table 1 diagnostics-11-00637-t001:** Descriptive analysis of the sample (mean ± standard deviation).

Age	All (N = 316)			
	N	Weight (kg)	Height (cm)	BMI (kg/m^2^)
6	36	23.7 ± 4.7	119.2 ± 5	16.5 ± 2.13
7	52	27.7 ± 6.4	123.6 ± 6.3	17.9 ± 2.7
8	40	31.2 ± 5.7	129.3 ± 4.6	18.6 ± 3
9	31	35.3 ± 9.1	135.4 ± 7.1	19 ± 3.6
10	68	37.7 ± 8	143.3 ± 6.3	18.3 ± 3.1
11	56	42.9 ± 11.1	148.9 ± 7.8	19.1 ± 3.6
12	33	50.1 ± 10.9	153.1 ± 7.3	21.3 ± 4
All	316	35.6 ± 11.4	136.8 ± 5	18.6 ± 3.4
**Boys** (N = 158)				
6	20	24.7 ± 5	119.4 ± 5.1	17.2 ± 2.5 *
7	22	26.2 ± 4.9	123.2 ± 5	17 ± 2.5
8	21	32.3 ± 5.8	130.9 ± 3.7 *	18.8 ± 3.3
9	18	34.3 ± 7.7	134.7 ± 6.3	18.8 ± 34.3
10	32	38.2 ± 7.4	143.1 ± 5.5	18.5 ± 3.1
11	28	42.1 ± 11.5	147.9 ± 6.5	19.1 ± 4.2
12	17	47.7 ± 11.1	151.7 ± 9.1	20.7 ± 4
All	158	35.3 ± 10.8	136.5 ± 10.3	18.6 ± 3.4
**Girls** (N = 158)				
6	16	22.4 ± 3.5	118.9 ± 5.1	15.7 ± 1.4 *
7	30	28.9 ± 7.2	124 ± 7.1	18.5 ± 2.8
8	19	29.8 ± 5.4	127.5 ± 4.9 *	18.4 ± 2.8
9	13	36.6 ± 10.7	136.4 ± 8.3	19.4 ± 4
10	36	37.2 ± 8.6	143.5 ± 6.9	18 ± 3.2
11	28	43.7 ± 10.9	150 ± 8.9	19.1 ± 2.8
12	16	52.7 ± 10.3	154.6 ± 4.5	21.8 ± 4.1
All	158	35.9 ± 11.9	137.1 ± 10.3	18.6 ± 3.3

BMI: body mass index. t-test between sexes: * *p* < 0.05.

**Table 2 diagnostics-11-00637-t002:** Average acceleration during monopodal test with eyes open and eyes closed (mean ± standard deviation).

Age	Boys		Girls	
	Eyes Open	Eyes Closed	Eyes Open	Eyes Closed
**Vertical Axis**				
6	11 ± 10.4 * ^a^	15.7 ± 13.9 ^a^	4.3 ± 4.1 * ^b^	8.6 ± 7.6 ^b^
7	6.7 ± 5 ^aa^	10.7 ± 7.1 ^aa^	7 ± 5.7 ^b^	11.1 ± 9.5 ^b^
8	4.2 ± 4.1 * ^aa^	7.2 ± 6.4 * ^aa^	2.5 ± 1.9 * ^bbb^	3.1 ± 3.7 * ^bbb^
9	3.4 ± 3.1 ^aa^	7.5 ± 7.4 ^aa^	1.6 ± 1.3 ^b^	3.5 ± 3.5 ^b^
10	8.3 ± 5.3 **	10.1 ± 8.5 **	2 ± 1.2 ** ^b^	3.6 ± 2.3 ** ^b^
11	2 ± 1.3 ^aa^	4.8 ± 4 ^aa^	1.8 ± 0.6 ^bb^	3.2 ± 2.6 ^bb^
12	1.4 ± 0.8 ^aa^	4.2 ± 4 ^aa^	0.3 ± 0.1	3.8 ± 1.8
All	6.7 ± 4.9 *** ^aaa^	9.1 ± 8 *** ^aaa^	4.3 ± 2.1 *** ^bbb^	6.4 ± 4.9 *** ^bbb^
**Mediolateral Axis**				
6	19.7 ± 13.9 ^aaa^	27.2 ± 13.6 ^aaa^	13.2 ± 7.6 ^bb^	22.1 ± 13.6 ^bb^
7	14.3 ± 9.2 * ^aaa^	21.5 ± 8.8 * ^aaa^	10.2 ± 8.2 * ^bbb^	15.6 ± 0.1 * ^bbb^
8	10.2 ± 8.4 * ^aaa^	15.5 ± 9.1 * ^aaa^	4.7 ± 3.7 * ^bbb^	10.1 ± 4.9 * ^bbb^
9	8.8 ± 6.7 * ^aaa^	16.3 ± 8.3 * ^aaa^	4.1 ± 3.9 * ^bb^	9.5 ± 8.7 * ^bb^
10	7.8 ± 6.4 ** ^aaa^	12.9 ± 10.1 ** ^aaa^	2.7 ± 2.5 ** ^bbb^	6.9 ± 6.1 ** ^bbb^
11	4.9 ± 3.2 * ^aaa^	10.2 ± 6.6 *** ^aaa^	2.8 ± 1.3 * ^bbb^	4.3 ± 4.1 *** ^bbb^
12	1.7 ± 1.7 ^aaa^	8.3 ± 7.1 ^aaa^	1.9 ± 0.3 ^bb^	5.6 ± 4 ^bb^
All	9.6 ± 9.1 *** ^aaa^	15.7 ± 10.8 *** ^aaa^	6.2 ± 5.3 *** ^bbb^	10 ± 9.5 *** ^bbb^
**Anteroposterior Axis**				
6	13.1 ± 7.6 ^aaa^	18 ± 9.3 ^aaa^	9.2 ± 7.8 ^bb^	13.4 ± 11.5 ^bb^
7	10 ± 6 ^aa^	13.1 ± 6.9 ^aa^	7.8 ± 6.1 ^bbb^	11.7 ± 6.1 ^bbb^
8	7.4 ± 5 * ^aa^	10.7 ± 7.2 * ^aa^	4.3 ± 4 * ^bb^	6.6 ± 4.9 * ^bb^
9	6.3 ± 6 * ^aaa^	9.6 ± 6.7 * ^aaa^	2.1 ± 1.8 * ^b^	5 ± 4.6 * ^b^
10	6.3 ± 5.8 * ^aaa^	7.8 ± 7.2 * ^aaa^	3 ± 2.9 * ^bb^	5.9 ± 4.1 * ^bb^
11	4.6 ± 3.4 * ^aaa^	6.5 ± 4.3 * ^aaa^	2.7 ± 1.4 * ^bbb^	3.8 ± 3.8 * ^bbb^
12	1.4 ± 1.1 ^aa^	4.8 ± 4.7 ^aa^	1.9 ± 0.6 ^bb^	4.5 ± 3 ^bb^
All	6.6 ± 6.6 *** ^aaa^	9.9 ± 7.7 ** ^aaa^	5.2 ± 4.1 *** ^bbb^	6.9 ± 6.8 ** ^bbb^
**Root Mean Square**				
**6**	26.8 ± 18.5 ^aaa^	37.4 ± 19.3 ^aaa^	27.6 ± 18.9 ^bb^	27.6 ± 18.9 ^bb^
**7**	19.1 ± 11.5 ^aaa^	28.2 ± 11.3 ^aaa^	14.7 ± 11.6 ^bbb^	22.6 ± 14.9 ^bbb^
**8**	13.6 ± 10.2 * ^aaa^	20.8 ± 12.2 * ^aaa^	6.9 ± 5.6 * ^bbb^	12.9 ± 7.1 * ^bbb^
**9**	11.7 ± 9.5 * ^aaa^	21 ± 11.9 * ^aaa^	5 ± 4.4 * ^bb^	11.5 ± 10.2 * ^bb^
**10**	12.5 ± 10.7 ** ^aaa^	17.6 ± 15.8 ** ^aaa^	4.5 ± 4 ** ^bbb^	9.1 ± 8.3 ** ^bbb^
**11**	6.7 ± 5.2 * ^aaa^	13.2 ± 8.6 ** ^aaa^	4.1 ± 1.5 * ^bbb^	6.5 ± 6.3 ** ^bbb^
**12**	2.5 ± 2.3 ^aaa^	10.8 ± 9 ^aaa^	2.3 ± 1 ^bb^	7.4 ± 6.2 ^bb^
**All**	13.6 ± 12.2 *** ^aaa^	20.9 ± 15.2 *** ^aaa^	8.4 ± 7.8 *** ^bbb^	13.5 ± 13.1 *** ^bbb^

*t*-test between sexes: * *p* < 0.05; ** *p* < 0.01; *** *p* < 0.001. Comparison between eyes open and eyes closed for boys: ^a^
*p* < 0.05; ^aa^
*p* < 0.01; ^aaa^
*p* < 0.0001. Comparison between eyes open and eyes closed for girls: ^b^
*p* < 0.05; ^bb^
*p* < 0.01; ^bbb^
*p* < 0.0001.

**Table 3 diagnostics-11-00637-t003:** Maximum acceleration during monopodal test with eyes open and eyes closed (mean ± standard deviation).

Age	Boys		Girls	
	Eyes Open	Eyes Closed	Eyes Open	Eyes Closed
**Vertical Axis**				
6	64.1 ± 42.9 *	72.2 ± 40.9	38.4 ± 26.3 * ^b^	53.2 ± 36.5 ^b^
7	55.5 ± 33.6	65.7 ± 30.3	41.4 ± 36.6 ^b^	54.4 ± 34.8 ^b^
8	36.5 ± 29.1 ^aa^	50.1 ± 29.2 ^aa^	20.1 ± 19.4 ^bb^	35.8 ± 21.6 ^bb^
9	31.4 ± 22.6 * ^aa^	48.4 ± 30.4 ^aa^	16.5 ± 14.8* ^b^	31.7 ± 22.2 ^b^
10	36.3 ± 40.4 ** ^aa^	50.9 ± 41.8 ** ^aa^	16 ± 14.3 ** ^bb^	28 ± 23.5 ** ^bbb^
11	16.9 ± 19.1 ^aa^	33.5 ± 23.9 ^aa^	9.9 ± 8.3 ^bbb^	22.3 ± 21.6 ^bbb^
12	10.7 ± 9.9 ^aa^	36.9 ± 31.9 * ^aa^	2.5 ± 2.4 ^bb^	18.5 ± 15.9 * ^bb^
All	35.8 ± 35.3 *** ^aaa^	50.7 ± 35.3 *** ^aaa^	21.2 ± 19.9 *** ^bbb^	34.5 ± 29.4 *** ^bbb^
**Mediolateral Axis**				
6	74.2 ± 33.7 ^aa^	88.7 ± 32.2 ^aa^	63.5 ± 26.1 ^bb^	82.3 ± 33 ^bb^
7	65.9 ± 28.2 ^aa^	82 ± 22.1 * ^aaa^	56.3 ± 29.8	65.4 ± 27.8*
8	53 ± 32.2 *	67 ± 29.4	34 ± 15.4 * ^bbb^	54.4 ± 20.8 ^bbb^
9	52.8 ± 21.5 ** ^aa^	68.7 ± 22 * ^aa^	30.9 ± 15.6 ** ^bbb^	50.1 ± 22.2 * ^bbb^
10	38.6 ± 29 * ^aaa^	62.3 ± 35.9 ** ^aaa^	24.4 ± 17.8 * ^bbb^	42.5 ± 22.5 ** ^bbb^
11	32.1 ± 22.1 ** ^aaa^	53.4 ± 18 *** ^aaa^	16.6 ± 13.8 ** ^bbb^	30.0 ± 19.3 *** ^bbb^
12	19.5 ± 18.2 ^aa^	46.7 ± 34.2 ^aa^	10.6 ± 11.5 ^bb^	31.9 ± 21 ^bb^
All	47.2 ± 31.6 *** ^aaa^	66.5 ± 30.9 *** ^aaa^	33.3 ± 26.6 *** ^bbb^	49.6 ± 28.7 *** ^bbb^
**Anteroposterior Axis**				
6	63.9 ± 22.5 ^aa^	82.7 ± 29.5 ^aa^	57 ± 28.6 ^b^	69.8 ± 30.4 ^b^
7	57.3 ± 23.1	64.2 ± 26	45.2 ± 20.5 ^bbb^	61.7 ± 20.4 ^bbb^
8	48.3 ± 28.3	57.1 ± 27.4	34.7 ± 23.4 ^b^	43.8 ± 17.9 ^b^
9	44.4 ± 22.5 **	48.2 ± 21.7 *	22.5 ± 14.6 ** ^b^	31 ± 17.3* ^b^
10	34 ± 32.2 ^aa^	45.8 ± 29.7 ^aa^	28.4 ± 23.8 ^bb^	38.2 ± 24.9 ^bbb^
11	29.5 ± 29.4 * ^aa^	43.3 ± 18 ^aa^	17.4 ± 19.2 * ^bb^	32.4 ± 24.7 ^bb^
12	14.2 ± 13.1 ^a^	34.8 ± 31.5 ^a^	10.6 ± 8.3 ^bbb^	27.1 ± 20.5 ^bbb^
All	41.1 ± 29.9 ** ^aaa^	53.2 ± 29.5 ** ^aaa^	30.8 ± 25.2 ** ^bbb^	43.8 ± 26.6 ** ^bbb^
**Root Mean Square**				
6	120.1 ± 52.1 ^a^	138.1 ± 52.1 ^a^	95.7 ± 42 ^bbb^	123.8 ± 49.3 ^bbb^
7	107.0 ± 40.9 ^a^	126.7 ± 34 ^a^	87 ± 44.5 ^bb^	108.5 ± 40.6 ^bb^
8	83.2 ± 47.2 * ^a^	105 ± 40.8 * ^a^	54.8 ± 31.5 ** ^bb^	80.2 ± 30.5 * ^bb^
9	77.4 ± 34.9 ** ^aa^	99.1 ± 37.7 * ^aa^	43.5 ± 22.6 * ^bb^	68.4 ± 32.6 * ^bbb^
10	65.3 ± 56.6 * ^aaa^	95.2 ± 58.5 * ^aaa^	43.3 ± 32.2 * ^bbb^	65.4 ± 38.1 * ^bbb^
11	48.9 ± 38.8 * ^aa^	78.9 ± 28.7 ** ^aa^	27.7 ± 25.3 * ^bbb^	52.2 ± 34.1 ** ^bbb^
12	27.5 ± 24.5 ^aa^	70.9 ± 53.7 ^aa^	15.6 ± 14.4 ^bbb^	46.5 ± 32.2 ^bbb^
All	74.8 ± 52.3 *** ^aaa^	102.1 ± 49.7 *** ^aaa^	52.6 ± 41.9 *** ^bbb^	77.3 ± 44.8 *** ^bbb^

*t*-test between sexes: * *p* < 0.05; ** *p* < 0.01; *** *p* < 0.001. Comparison between eyes open and eyes closed for boys: ^a^
*p* < 0.05; ^aa^
*p* < 0.01; ^aaa^
*p* < 0.0001. Comparison between eyes open and eyes closed for girls: ^b^
*p* < 0.05; ^bb^
*p* < 0.01; ^bbb^
*p* < 0.0001.

**Table 4 diagnostics-11-00637-t004:** Models of logistic regression for sex adjusted for postural control root mean square accelerometric values and age.

	ME	OR	SE	CI 95%
**Monopodal Balance with Eyes Open Test** (g)				
Age	−0.051	0.816 **	0.06	0.705–0.945
Average root mean square	−0.017	0.935 ***	0.014	0.907–0.963
Constant		12.077 ***	9.403	2.626–55.546
Age	−0.053	0.807 **	0.062	0.695–0.937
Maximum root mean square	−0.004	0.985 ***	0.003	0.979–0.991
Constant		18.066 ***	15.113	3.506–93.095
**Monopodal Balance with Eyes Closed Test** (g)				
Age	−0.052	0.813 **	0.06	0.703–0.942
Average root mean square	−0.014	0.946 ***	0.011	0.001–0.926
Constant		16.435 ***	13.073	3.457–78.14
Age	−0.048	0.824 **	0.06	0.714–0.952
Maximum root mean square	−0.004	0.985 ***	0.003	0.979–0.991
Constant		22.555 ***	19.217	4.246–119.806

ME: marginal effects after logit in percentage; OR: odds ratio; SE: standard error; CI: confidence interval; RMS: root mean square. * *p* < 0.05; ** *p* < 0.01; *** *p* < 0.001.

## Data Availability

The data presented in this study are available on request from the corresponding author.

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
