# Peer review of "Influence of Visual Information and Sex on Postural Control in Children Aged 6–12 Years Assessed with Accelerometric Technology"

_diagnostics, 2021, doi:10.3390/diagnostics11040637_

Round 1
Reviewer 1 Report
This reviewer is not confident whether the statistical analyses employed in the present manuscript are indeed appropriate. It seems that the probability of committing Type I and Type II errors are increased due to multiple comparisons. The authors have changed their analyses in this revision; nevertheless I still think that dividing the analyses by sex and by age group might not be appropriate. The results do not make clear which test were performed for each assumption: degrees of freedom and T/F values are missing (only p values and effect sizes are reported in the revised version, which is still not enough). I therefore recommend the editors to send the present manuscript to be seen by an expert statistician before publication.
Showing the results with figures rather than tables would be preferable.
English revision is needed throughout.
Reviewer 2 Report
The revision is fine and well justified.
This manuscript is a resubmission of an earlier submission. The following is a list of the peer review reports and author responses from that submission.
Round 1
Reviewer 1 Report
The topic is quite interesting.
The introduction part is rather weak and lacks of comprehensive review on the topic. For example, line 43, ‘age’ is a very influential factor….but there is no further justification in relation with age and other postural control measure which affect children’s behavior/physical development. More related literature on postural control should be included in the introduction so as to provide general background for the readers.
2.1 Design and sample – the background of the school children should also provide because environmental and social factors might contribute to the children’s physical development.
The discussion should provide some practical suggestions for the on-field educators based on the present finding. For example, line 167, ‘vision’ is important….but there is no practical recommendation on how to make it better.
Line 184 paragraph, the balance between sexes were greater…. It needs to add more literature to explain such differences across the age group from 6 to 12 years.
In the conclusion, line 230 “intervention”, owing to the fact that various environmental, social, parental and cultural factors can also influence the children’s physical development, the researcher should provide practical on-field suggestions for the readers. E.g. school-based intervention program or after school exercise program, etc.
Reviewer 2 Report
This interesting manuscript aims to explore postural stability in children aged 6-12 years during static single-leg support (the influence of vision availability was also assessed). The sample size is quite large and the paper is well written, despite some grammar and syntax mistakes.
Unfortunately, there are major points that, in my opinion, preclude publication of this manuscript in its present form:
1) The authors must justify the choice for measuring acceleration rather than center of mass / center of pressure displacements (which are the “gold standards” for postural assessments). Additionally, the rationale for choosing a single-leg task (rather than a more “natural” bipedal quiet stance) is also needed.
2) Statistical analyses are not appropriate. The authors used Student’s T-test to identify statistical differences between sexes. However, multiple comparisons were also made (for each of the 7 different ages), so that the probability of committing Type I and Type II errors is quite high. An adjustment for multiple comparisons is needed.
3) The correlation analyses / regression model is not well justified on the basis of the primary and secondary aims of this investigation.
4) The presentation of the results as tables with mean ± SD vales was not a good choice. Figures (bars or symbols with error bars) would be preferable so as to allow visualization of the results as a function of sex and age.
5) Giving P values are not enough. T values, degrees of freedom (considering the correct adjustments for multiple comparisons, as commented above) and measures of effect sizes are necessary.
6) It cannot be inferred that girls “prefer” to use somathestetic information as this was not assessed in the present study. Sensory manipulation involved eyes open / eyes closed conditions only. There are much more complex sensory and biomechanical factors that influence postural control than simply vision vs somathestetic information. Finally, “prefer” is not a good term here, as it implies some “cognitive-based preference”, which is clearly not the case.